# Assessment of Geo-Kompsat-2A Atmospheric Motion Vector Data and Its Assimilation Impact in the GEOS Atmospheric Data Assimilation System

Eunhee Lee [1,2,3], Ricardo Todling [3], Bryan M. Karpowicz [3,4,5], Jianjun Jin [3,6], Akira Sewnath [3,6] and Seon Ki Park [2,7,*]

1. Korea Meteorological Administration, Seoul 07062, Korea
2. Department of Environmental Science and Engineering, Ewha Womans University, Seoul 03760, Korea
3. Global Modeling and Assimilation Office, NASA Goddard Space Flight Center, Greenbelt, MD 20771, USA
4. Goddard Earth Sciences and Technology and Research II, Greenbelt, MD 20768, USA
5. Earth and Space Research Administration, University of Maryland Baltimore County, Baltimore, MD 21227, USA
6. Science Systems and Applications, Inc., Lanham, MD 20706, USA
7. Department of Climate and Energy Systems Engineering, Ewha Womans University, Seoul 03760, Korea
* Correspondence: spark@ewha.ac.kr; Tel.:+82-2-3277-3331

**Abstract:** Korea's second geostationary meteorological satellite, Geo-Kompsat-2A (Geostationary-Korean Multi-Purpose Satellite-2A, GK2A), was successfully launched on 4 December 2018. GK2A generates Atmospheric Motion Vectors (AMVs) every 10 min in the full disk area. This data has been disseminated via Global Telecommunication System (GTS) since 25 October 2019. This article evaluates the quality of GK2A AMVs in the Goddard Earth Observing System (GEOS) atmospheric data assimilation system (ADAS). The data show slow wind speed biases at 200–300 hPa and 600–800 hPa in the northern and southern hemispheres. These biases are caused by observation height assignment errors near jet streams. The Equivalent Blackbody Temperature (EBBT) method of GK2A tends to assign clouds at higher altitude, which mainly causes slow wind speed biases, especially in the lower atmosphere. The IR/WV intercept method of GK2A assigns clouds slightly lower in the atmospheric layers below the altitude of 400 hPa, which causes positive biases. Quality control (QC) criteria to select the most suitable GK2A AMV data for assimilation are presented based on these quality assessments. A new QC criterion utilizing height errors within the GEOS ADAS is introduced to exclude data with slow wind speed biases and large errors. GEOS forecast accuracy is slightly improved after assimilating GK2A AMVs along with other conventional, radiance, and satellite winds which include AMVs made by the Himawari-8 satellite in nearly the same observational area of GK2A. Additionally, the present work shows that GEOS forecasts can be significantly improved, especially in the tropics and southern hemisphere after assimilating GK2A data in the absence of Himawari-8 AMVs. This study demonstrates that GK2A AMV data is a valuable data source to enhance the robustness of GEOS ADAS.

**Keywords:** Geo-Kompsat-2A (GK2A); atmospheric motion vector (AMV); height error; quality control; assimilation

## 1. Introduction

The new generation of satellites equipped with advanced imagers, such as Geostationary Operational Environmental Satellite (GOES)-16/17, Himawari-8, Fengyun-4A, Geo-Kompsat-2A, and Meteosat Third Generation (MTG) are expected to provide high resolution and high-quality Atmospheric Motion Vectors (AMVs). Recent studies have shown AMVs from these new satellites have significantly increased observation density, decreased speed bias, and assured better quality when compared to AMVs from previous

satellite generations [1–4]. In addition, various Observing System Experiments (OSEs) show positive impact of these new high-resolution satellite AMVs even in the presence of abundant satellite radiance observations in global and regional weather forecasts [1–3,5–7] and tropical cyclone and hurricane forecasts [8–10].

Satellite AMVs provide wind information with dense spatial and temporal resolution, especially over wind-data scarce areas such as oceans, the southern hemisphere, and polar regions. In particular, AMVs derived from geostationary satellites have high temporal resolution, providing continuous information of fast-changing weather phenomena such as convection. Thus, AMVs have become an indispensable part of improving numerical weather prediction (NWP) accuracy since the early 1970s when they began to be derived from consecutive satellite images by tracking moving objects such as clouds and water vapor [11–14]. AMV data from geostationary and polar orbit satellites are assimilated in many operational NWP systems with demonstrably positive impact to model predictions [15–20].

The Geo-Kompsat-2A (GK2A) satellite, one of the new generation of geostationary satellites, is located at 128.2°E, and provides high spatial and temporal AMVs above the Indian Ocean and Asia-Pacific where wind observations are scarce. However, GK2A AMVs are not yet used in operational NWP centers, except for the Korea Meteorological Administration (KMA). They are, however, being evaluated for their impact by multiple operational organizations. The main purpose of this paper is to evaluate the impact of GK2A AMVs on NASA's Goddard Earth Observing System (GEOS) NWP system.

Quality control (QC) is essential to bring new observations into the data assimilation system and ensure a positive impact in regional and global data assimilation [21]. A critical component in the QC is selecting the optimal data for the NWP by removing observations with large uncertainties. AMV observation uncertainties can be estimated from two types of errors which occur during vector tracking and height assignment processes. Satellite retrieval algorithms assume a single-layer cloud, which causes AMV height assignment uncertainty [22,23]. This is because radiance observed by a satellite imager is emitted from various atmospheric layers, not from a specific single level. The AMV height assignment uncertainty is greater in areas where there is large vertical wind shear [24]. The height assignment is the dominant factor contributing up to 70% of total AMV uncertainty [25].

This study aims to evaluate the quality of GK2A AMV data within the context of the NASA Global Modeling and Assimilation Office (GMAO), quasi-operational, GEOS atmospheric data assimilation system (ADAS). We present QC criteria for optimal use of GK2A AMVs and verify their effectiveness in the GEOS ADAS. Section 2 provides an overview of GK2A AMV data and the methodology used in this study. Data quality analysis of the GK2A AMVs based on the height error is also presented. Based on this analysis, QC criteria and experimental designs are constructed to test the effectiveness of the data in the global NWP system. Results are presented in Section 3, followed by a summary and discussion of findings in Section 4.

## 2. Data and Methods

### 2.1. Geo-Kompsat-2A AMV Datasets

The GK2A satellite, the second Korean geostationary meteorological satellite, was successfully launched on 5 December 2018, taking over the mission of the Communication, Ocean, and Meteorological Satellite (COMS), South Korea's first meteorological satellite launched in 2011. The Advanced Meteorological Imager (AMI) on GK2A is more advanced than the COMS's Meteorological Imager (MI) with higher radiometric, spectral, and spatial resolution [26]. The spatial resolution of AMI is twice as high as that of MI, and temporal resolution is 18 times higher because AMI produces a full-disk image every 10 min while MI produces one every 3 h. AMVs are provided by both COMS and GK2A, and GK2A AMVs are denser and better quality than COMS AMVs [4]. Lee et al. [27] showed overall forecast improvement by assimilating COMS AMVs into the KMA global NWP system, and those data have been operationally used in the system since December 2011. GK2A AMVs

replaced COMS AMVs in 2020 when the COMS mission was completed, and continues to be used in the current operational KMA NWP system.

The present study uses the GK2A AMV product from December 2020 to January 2021. The dataset is generated and disseminated in BUFR format by KMA National Meteorological Satellite Center (NMSC), and has been disseminated via the Global Telecommunication System (GTS) since 25 October 2019. The data are converted to the Prepared Binary Universal Form for the Representation of Meteorological Data (PREPBUFR) format for ingestion in the GEOS ADAS. The GK2A AMVs are derived from three consecutive full disk images in 10-min intervals. The GK2A AMVs are retrieved individually from the visible (VIS) channel at 0.64 μm, short-wave infrared (SWIR) at 3.83 μm, three water vapor (WV) channels at 6.21 μm, 6.94 μm and 7.32 μm, and two infrared (IR) channels at 10.35 μm and 11.23 μm. In this study, since winds from the SWIR channel have relatively small amount of data and poor quality compared to those from other channels, AMV data from six channels excluding SWIR are used. The GK2A AMI channel characteristics used in the AMV derivation are shown in Table 1, which shows similar radiometric, spectral, and spatial resolution to the Advanced Himawari Imager (AHI) of Himawari-8 (H-8) as well. The original spatial resolution of the VIS channel is 500 m and the resolution of the other channels is 2 km. The AMV spatial resolutions are 48×48 pixels (24 km × 24 km) and 16×16 pixels (32 km × 32 km) for the VIS channel and the other channels, respectively. They are determined by sensitivity experiments which search the optimal target box. Wind speed and direction are calculated by using the cross-correlation coefficient (CCC) [23] and the height of each vector is assigned by using CCC, equivalent blackbody temperature (EBBT) [28], IR/WV intercept [29], and $CO_2$ slicing [30] methods for cloudy targets. In contrast, normalized total contribution (NTC) and normalized total cumulative contribution (NTCC) [4] are used to assign the height of each vector for clear-air targets. AMVs derived using clear-air are not assimilated in this study due to their poor quality. Details on GK2A AMV derivation can be found in Oh et al. [4].

**Table 1.** Comparison between characteristics of the GK2A AMI and H-8 AHI used to derive AMV product.

| | | | AMI | | AHI | |
|---|---|---|---|---|---|---|
| Channel | Band | Spatial Resolution (km) | Central Wavelength (μm) | Band Width (μm) | Central Wavelength (μm) | Band Width (μm) |
| VIS | 3 | 0.5 | 0.639 | 0.081 | 0.0645 | 0.03 |
| SWIR | 7 | 2.0 | 3.83 | 0.19 | 3.85 | 0.22 |
| WV | 8 | 2.0 | 6.21 | 0.84 | 6.25 | 0.12 |
| | 9 | 2.0 | 6.94 | 0.40 | 6.95 | 0.12 |
| | 10 | 2.0 | 7.33 | 0.18 | 7.35 | 0.17 |
| IR | 13 | 2.0 | 10.35 | 0.47 | 10.45 | 0.30 |
| | 14 | 2.0 | 11.23 | 0.66 | 11.20 | 0.20 |

*2.2. Initial Quality Assessment of GK2A AMV Data*

All AMV products contain a Quality Indicator (QI) which represents statistical quality information of the derived AMVs. QI is introduced to accurately assess the quality and representativeness of individual wind vectors and to provide this information to NWP centers. It is calculated from the weighted average of five components: speed, direction, vector, forecast and spatial consistencies [31,32]. This is usually used as a QC criterion to screen poor quality AMV data before assimilation in NWP systems. GK2A AMVs are provided with three types of QI: forecast dependent QI (QIFY), forecast independent QI (QIFN) and common QI. The QIFN has a zero-weight value for forecast consistency.

Figure 1 shows dependence of Root Mean Square Vector Differences (RMSVD) on QIFN and QIFY values for different channels. In QIFY, there is a consistent pattern of decreasing RMSVD as the QIFY value increases when QIFY is greater than 75. In contrast,

QIFN does not show a pattern of decreasing RMSVD along with increasing QIFN values. As a result, it is difficult to establish a QIFN threshold for screening. Therefore, unlike other AMV data which are screened by QIFN in ADAS, GK2A is screened by QIFY in this study.

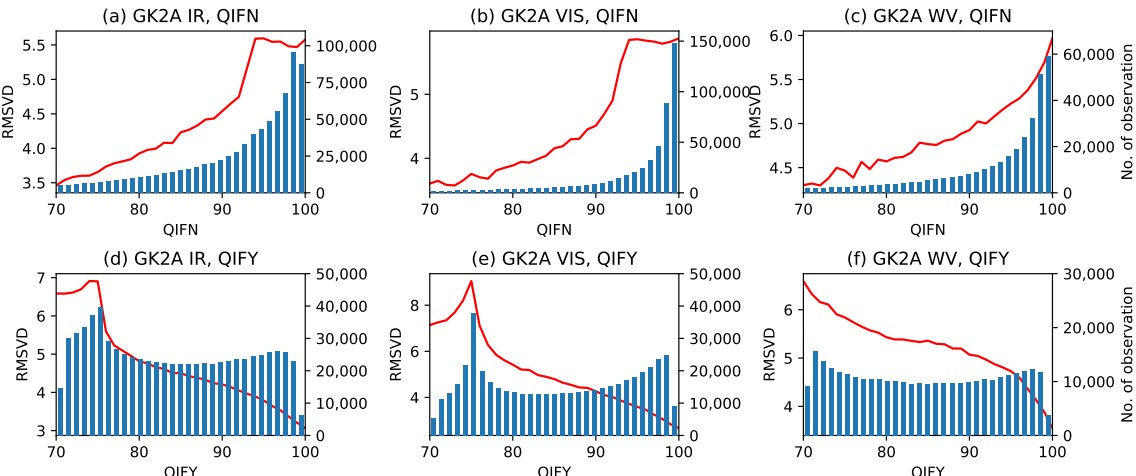

**Figure 1.** Relationship between Root Mean Square Vector Differences (RMSVD) and forecast independent QI (QIFN, **a–c**), forecast dependent QI (QIFY, **d–f**) for the IR (**a,d**), VIS (**b,e**) and WV (**c,f**) channel of GK2A during winter (December 2020). Red lines indicate RMSVD values and blue bars show the numbers of AMVs assigned to each QI.

Collocated in situ measurements such as radiosondes and wind profilers are commonly used to verify the quality of AMVs, and collocated cloud observations from lidar on satellites are used to assess height assignment of AMVs. However, only limited AMV data can be evaluated in this way due to insufficient spatial and temporal coverage of collocated measurements. Since the method of comparing and verifying AMVs with short-term forecast model is applicable to all AMV observations, the error characteristics of AMVs in regards to each satellite, channel, and height assignment method can be easily investigated. It is important to note that statistics from observation minus background (O–B) includes contributions from errors in the model background. In this study, O–B statistics from the GEOS system are used to analyze the quality of GK2A AMV data in the full disk area and to identify detailed error characteristics.

GK2A is positioned at 128.2°E and H-8 is located at 140°E, so the observation coverages of the two satellites overlap considerably (Figure 2). The AHI on H-8 has similar radiometric, spectral and spatial resolution to the AMI on GK2A, and the channels used to derive AMVs on both satellites are almost identical as shown in Table 1. In a previous inter-comparison study of AMVs from six organizations, H-8 AMVs produced by the Japan Meteorological Agency (JMA) show better statistics than AMV products derived by other organizations including KMA due to its new height assignment method, the optimal estimation method. However, this might indicate that the optimal estimation method has a stronger dependency on the NWP model background [33]. Since H-8 AMVs are already assimilated in the current GEOS ADAS, it is necessary to compare the quality of GK2A AMVs with H-8 AMVs before assimilating GK2A AMVs in the system.

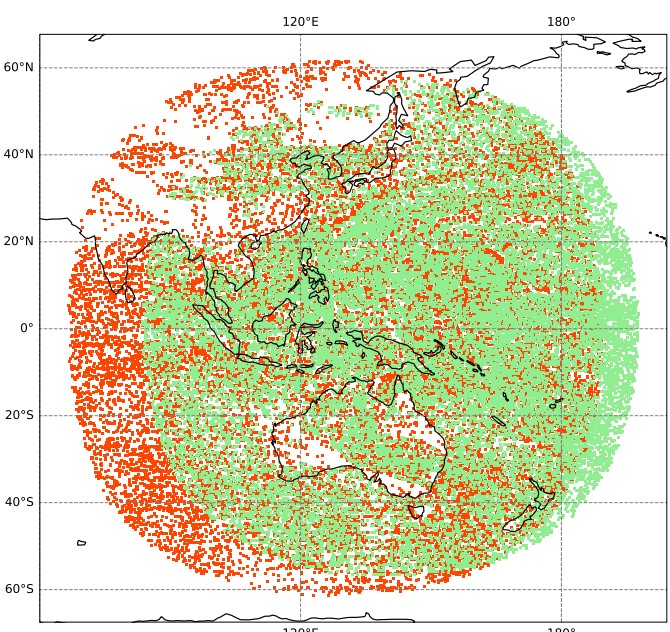

**Figure 2.** The spatial distribution of GK2A AMVs (orange) and H-8 AMVs (green) from GEOS ADAS on 1 December 2020.

Figure 3 shows the horizontal RMSVD and wind speed bias of GK2A (left) and H-8 (right) AMVs upper-level winds using GEOS model background winds in December 2020. The H-8 winds have a slightly faster wind speed bias in the tropics, while the GK2A IR winds show relatively high RMSVD and slower wind speed bias in the extratropics. The large wind-speed errors are likely due to a height assignment error and high vertical wind shear above Asia and Australia where strong jet streams exist in the upper troposphere [25,34,35].

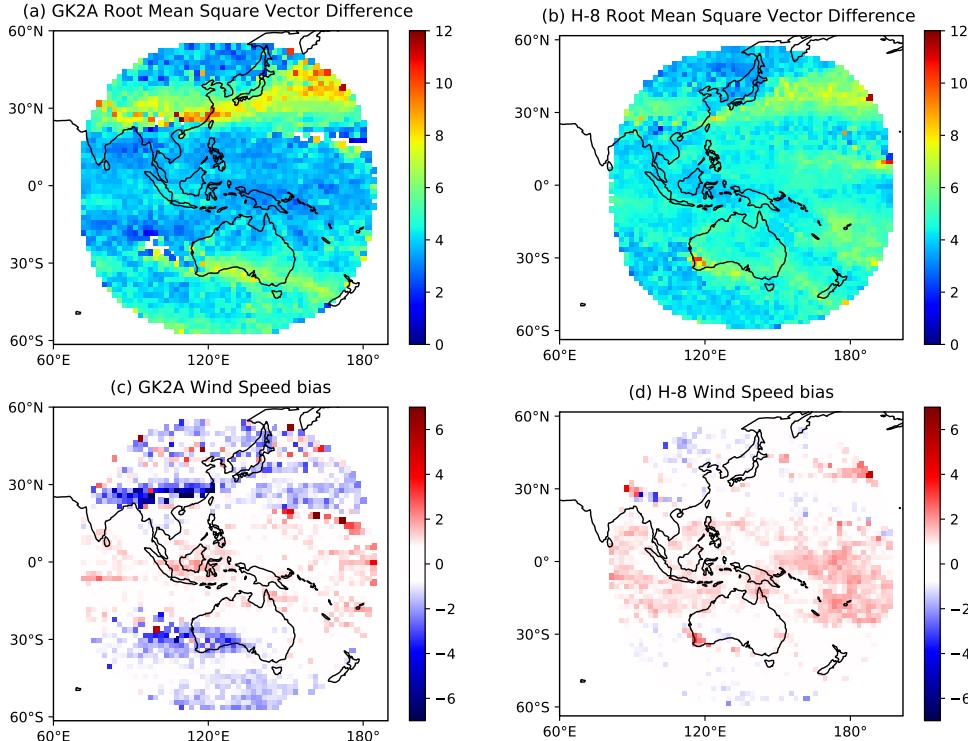

**Figure 3.** RMSVD (**a**,**b**) and wind speed bias (**c**,**d**) for IR winds at high level (<400 hPa) using winter data with GK2A data (**a**,**c**) and H-8 data (**b**,**d**).

Figure 4 shows mean wind speed biases at each pressure level of GK2A and H-8 IR winds. It confirms that GK2A has larger negative wind speed biases than H-8 not only in the upper layers but also in the lower layers between 600 and 800 hPa in the extratropics. These negative wind speed biases are dominant in the extratropical middle and upper troposphere while positive wind speed biases occur in the free troposphere in the tropics. GK2A AMVs have overall larger speed biases than H-8 AMVs in the extratropics while H-8 AMVs have larger speed biases than GK2A AMVs in the tropics, and the cause is analyzed in Section 2.3.

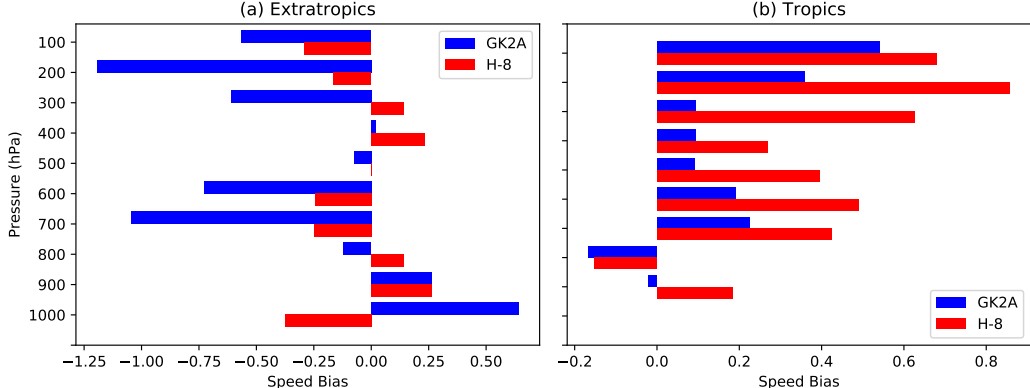

**Figure 4.** Speed bias of the GK2A (blue) and H-8 (red) AMVs in the extratropics (**a**) and tropics (**b**).

## 2.3. Height Assignment Error

The height assignment error is estimated by comparing the assigned pressure with the model best-fit pressure in many studies [24,33,35–37]. The best-fit pressure is defined as the pressure with the smallest vector difference between the observation and the model background winds. This method is useful to assess height assignment errors because the best-fit pressure can be defined for most AMVs, unlike the method to verify AMVs heights against co-located observations which is limited. However, there are uncertainties due to errors in NWP models and limitations in model vertical resolution.

The main difference between the GK2A and the H-8 AMVs retrieval algorithms is the height assignment method. The height of GK2A AMVs are assigned using EBBT or IR/WV intercept method while H-8 AMVs are derived using an optimal estimation method. For this reason, error characteristics generated by the height assignment method of GK2A are examined using the statistics of best-fit pressure from the GEOS ADAS. The height assignment error ($\sigma_p$) is estimated by the standard deviation of the assigned pressure of AMV minus the best-fit pressure. Figure 5 shows mean difference of the assigned pressure of GK2A IR winds minus the model best-fit pressure (hereafter referred to as PD) in December 2020. AMVs derived by the EBBT height assignment method have overall negative PD biases at all altitudes and large negative PD biases in the lower atmosphere between 600 and 800 hPa. In other words, observations are assigned at higher altitudes than the model best-fit pressure level. Since wind speed generally increases with altitude, assigning a higher than actual altitude increases the probability of negative wind speed bias. However, Figure 6 shows that the IR/WV intercept height assignment method tends to produce positive PD biases at an altitude below 400 hPa, which results in a positive wind speed biases between 400 and 600 hPa.

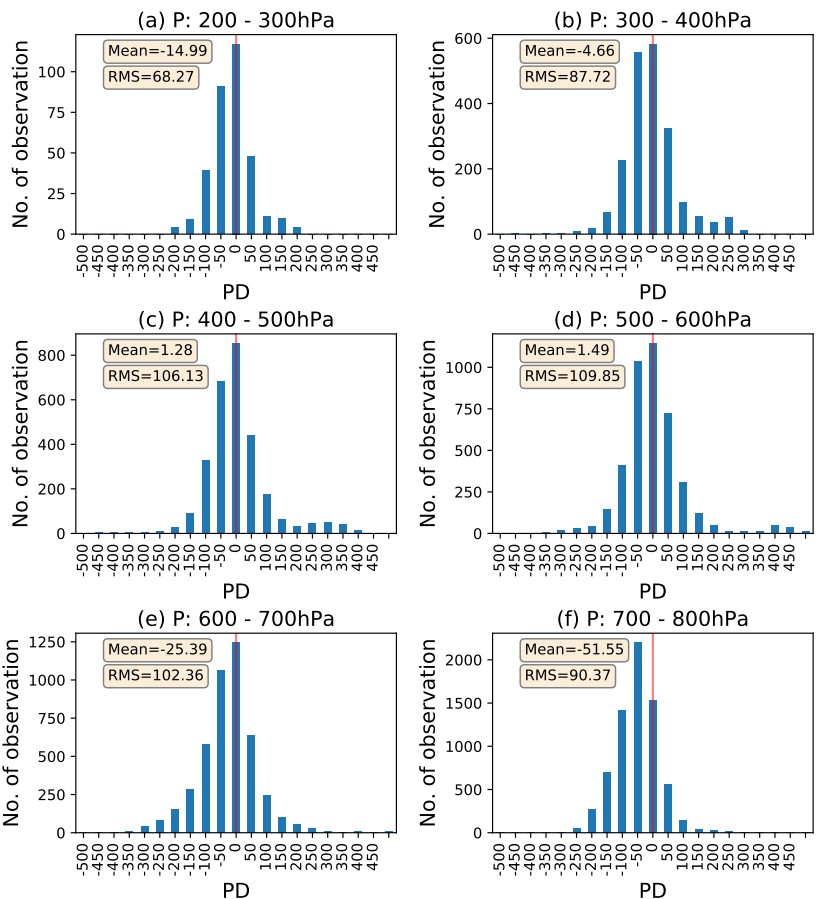

**Figure 5.** Histograms of PD for GK2A IR winds derived using the EBBT height assignment method at each pressure band during winter. Mean and RMS values of PD are presented. The red lines indicate that PD is zero.

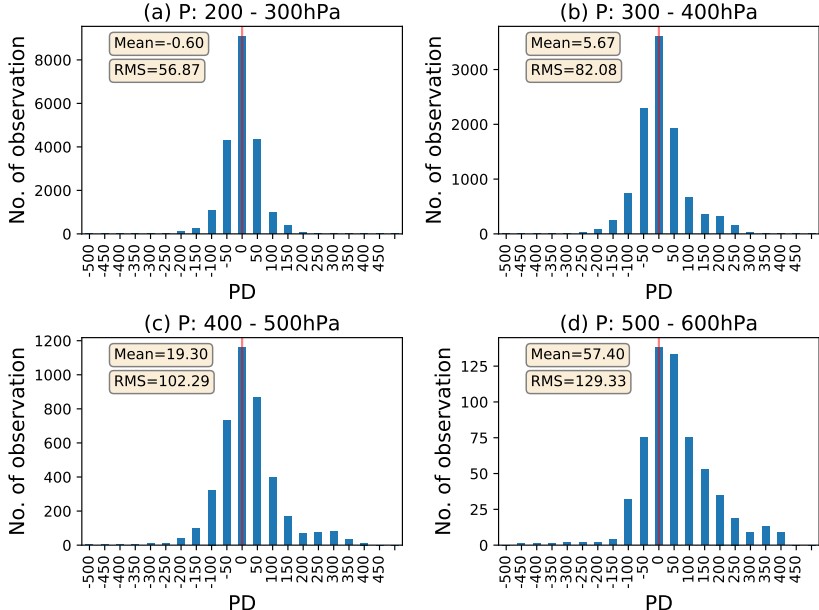

**Figure 6.** Same as Figure 5 except for GK2A IR winds derived using IR/WV intercept. Due to the characteristics of the IR/WV intercept, data exist only at altitude above the 600 hPa altitude.

Figure 7 displays scatter plots of PD values and wind speed difference between the observation and the model background winds (hereafter referred to as WSD). Overall, PD and WSD has a strong correlation. Positive PD values are associated with positive (fast) wind speed biases, and negative PD values are associated with negative (slow) wind speed biases. In the lower layer (600–800 hPa), there are notably more negative PD than positive PD values. Therefore, the EBBT method tends to assign clouds high, which causes slow wind speed biases, as shown in both Figures 4 and 5. The WV and VIS winds of GK2A show almost identical features as the IR winds (not shown).

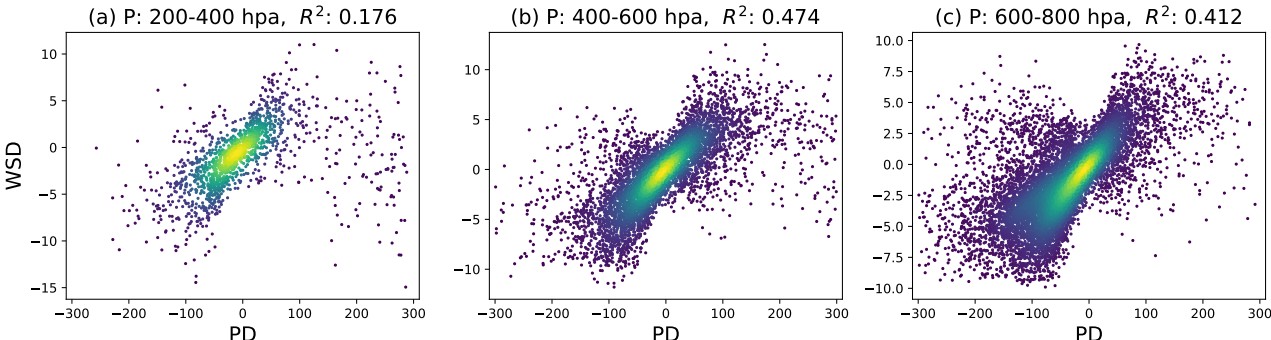

**Figure 7.** Scatterplots showing the relationship between PD (hPa) and WSD (m/s) of GK2A IR winds derived using the EBBT method in each atmospheric layer at (**a**) 200–400 hPa, (**b**) 400–600 hPa and (**c**) 600–800 hPa in the extratropics. QIFY threshold of 90 is applied.

The AMV height assignment errors and speed biases are greater in areas with large vertical wind shear. Figure 8 shows the zonal plot of wind shear and PD, speed bias and RMSVD of GK2A IR winds and H-8 IR winds. The vertical wind shear is calculated using the vertical change in wind speed as follows to simply analyze the quantitative correlation between wind speed bias and vertical wind shear:

$$\left| \frac{d|\vec{V}|}{dz} \right| \approx \left| \frac{\sqrt{u_{i+1}^2 + v_{i+1}^2} - \sqrt{u_i^2 + v_i^2}}{Z_{i+1} - Z_i} \right| \tag{1}$$

where $u$ and $v$ are the zonal and meridional components, $i$ is the background level, and $Z$ is altitude.

GK2A IR winds have a negative PD at the pressure level of 600–800 hPa, which is consistent with the region with a slow wind speed bias except for the tropics region, as shown in Figures 4 and 5. In addition, it can be seen that the speed bias and RMSVD are large in the region where the strong vertical wind shear and the PD are located together. H-8 AMVs, which have similar observational coverage to those of GK2A, show some weak negative PD near the jet stream and at the low level, but do not show as clear a negative wind speed bias as the GK2A (see Figure 8).

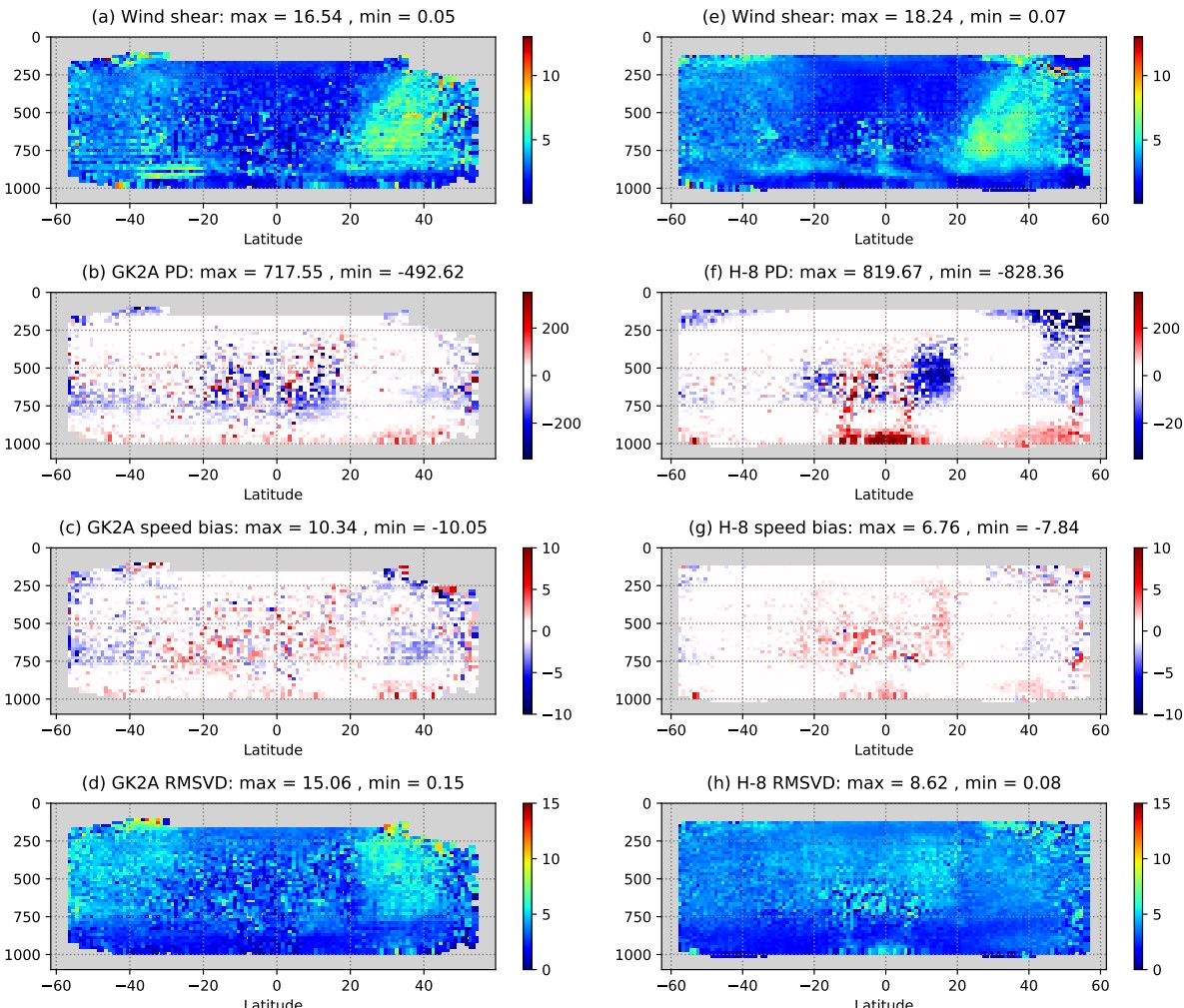

**Figure 8.** Zonal mean latitude-pressure cross-sections of wind shear, pressure differences, speed biases and the RMSVD for GK2A IR winds (**a–d**, left) and H-8 IR winds (**e–h**, right) during December 2020. Data filtered for QIFY < 90, SAZA > 64° (GK2A) or QIFN < 85, SAZA > 68° (H-8).

### 2.4. Quality Control

This work introduces the ability to assimilate GK2A AMVs with new QC criterion in the analysis component of GEOS ADAS. The GEOS ADAS integrates the GEOS Atmospheric Global Climate Model (AGCM, [38]) with the Gridpoint Statistical Interpolation (GSI, [39–41]) atmospheric analysis. The current version of GEOS ADAS uses 72 vertical levels with a model top at 0.01 hPa, and adopts a hybrid four dimensional ensemble-variational (4D-EnVar) configuration of GSI [42] and a 4D Incremental Analysis Update [43] approach to constrain the forecasting model integration.

Based on the quality assessment results of GK2A AMVs in this study and the channel characteristics of the AMVs from previous studies [6,34] and long term monitoring, the following QC criteria are implemented in GSI:

1.  Remove all the wind data above 150 hPa or the tropopause;
2.  Remove all the data from the surface to 950 hPa;
3.  Remove all the data with QIFY smaller than 90 and with satellite zenith angle (SAZA) larger than 64°;
4.  For the VIS winds, reject data above 700 hPa;
5.  For the WV winds, reject data below 400 hPa;
6.  For the IR winds, reject data over land if lat > 20°N;

7.  Reject IR and VIS winds derived using the EBBT method between 600 hPa and 800 hPa in the Southern Hemisphere to remove slow wind speed bias;
8.  Reject IR winds derived using IR/WV intercept below 400 hPa;
9.  A strict gross error check to eliminate the observation outside of tolerances from the model wind vector;
10. A height error check to eliminate the observation with large wind speed bias and RMSVD.

The QC criteria from first to sixth presented above are generally applied to all AMVs, and the thresholds of QIFY and SAZA are presented for GK2A AMVs through quality assessment in this study.

The seventh criterion is imposed to remove the prominent speed and height biases of GK2A AMVs since the variational analysis assumes unbiased observations.

The gross error check is already implemented in GSI, which eliminates observations that differ significantly from background field. It implements the follow bound:

$$\frac{\sqrt{(u_o - u_b)^2 + (v_o - v_b)^2}}{\sigma_o} > g \tag{2}$$

where $u_o$ and $v_o$ are the wind components of observation, $u_b$ and $v_b$ are the wind component of model background interpolated to observation location, $\sigma_o$ is the observation error, and $g$ is a global constant applied to each satellite and instrument type. In this study, we set the initial $g$ value for GK2A AMVs to 2.5 which is identical to the other geostationary satellite winds in the GEOS ADAS and tightened the (O–B) tolerances by a factor of 0.5 from the initial $g$ value for observations with slow speed bias.

A height error check is introduced in this work to handle GK2A AMVs. The height error $E_h$, $u/v$ error due to height assignment, can be calculated using the following weighted average and probability density function [24]:

$$E_h = \sqrt{\frac{\sum W_i (V_i - V_o)^2}{\sum W_i}} \tag{3}$$

$$W_i = \exp\left(\frac{-(P_i - P_o)^2}{2\sigma_p^2}\right) \times dP_i \tag{4}$$

where $V_i$ is the wind vector on model level $i$, $V_o$ is the wind vector at observation location, $P_i$ is the pressure on model level $i$, $\sum$ is the sum of all model levels below 100 hPa, $P_o$ is the pressure at observation location, $\sigma_p$ is the error in height assignment and $dP_i$ is the model layer thickness. $\sigma_p$ is defined through long-term monitoring of the difference between the assigned AMV pressure and the model best-fit pressure. These equations assume a Gaussian distribution of height assignment error, and there is a good indication from Figures 5 and 6 that this is a reasonable assumption. Equation (3) can be used to assess the uncertainty of AMV from height assignment error considering vertical wind shear. In order to remove the slow wind speed bias and large RMSVD caused by the height assignment error, AMVs are rejected if the height error is greater than 6 m/s.

### 2.5. Observing System Experiment Design

Observing System Experiments (OSEs) are conducted to investigate the impact of GK2A AMVs on model forecasts when assimilated together with or in the absence of H-8 AMVs. Table 2 lists the experiments evaluated in the remaining part of this study. The control experiment (CNTL) assimilates all observations typically used in the quasi-operational GEOS ADAS, including AMV observations from five geostationary, namely, GOES-16, GOES-17, Meteosat-8, Meteosat-11, Himawari-8, and six polar orbiting satellites, namely, Aqua, Terra, N-18, N-19, MetOp-A, MetOp-B. The substantial overlap between H-8 and GK2A observation areas (see Figure 2) motivates two experiments aimed at evaluating

the effectiveness of GK2A AMVs as a potential replacement for H-8 AMVs. Experiment NOH8 removes H-8 AMVs from CNTL and serves as a baseline for various comparisons. In this sense, CNTL represents an attempt to improve GEOS ADAS by adding H-8 AMVs to the data denial experiment NOH8. The experiment named GK2AHE adds GK2A AMVs to NOH8, using the QC criteria 1-10. Given that H-8 AMVs are missing in GK2AHE, the effectiveness of GK2A AMVs to serve as a replacement to H-8 AMVs can be derived by comparing GK2AHE with CNTL. Ultimately, experiment GK2AH8 adds GK2A AMVs to the current configuration of GEOS ADAS (CNTL).

**Table 2.** The list of experiments.

| Experiments | Assimilated Observations |
| --- | --- |
| CNTL | All conventional + satellite observations |
| NOH8 | Remove H-8 AMVs from CNTL (baseline) |
| GK2AHE | Add GK2A AMVs to NOH8 |
| GK2AH8 | Add GK2A AMVs to CNTL |

All experiments are based on the current version of GEOS ADAS used in the GMAO Forward Processing (near real-time) quasi-operational system. The experiments are performed in the so-called pre-parallel test configuration of GEOS ADAS which runs at a down-graded cubed-sphere horizontal resolution of C360 (approximately 25 km), with the full 72 vertical levels, and make use of the 6-hour hybrid 4D-EnVar strategy. All experiments in the present study "replay" the ensemble perturbations required by the hybrid deterministic analysis (see [42]). That is to say, the ensemble perturbations are derived from the CNTL experiment.

## 3. Results

### 3.1. Analysis Impact

Figure 9 shows zonally-averaged wind speed bias and RMSVD of the GK2A and H-8 IR winds assimilated in the experiment GK2AHE and CNTL that passed QC, respectively. Similar to the initial quality assessment results, the GK2A AMVs have a relatively large RMSVD and speed bias in the northern hemisphere while the H-8 AMVs have a relatively large RMSVD and a positive wind speed bias in the tropics. As a result of applying the QC criteria presented in the previous section, the systematic slow speed bias in GK2A is removed in tropics and in the lower extratropical layers. Total number of H-8 AMVs and GK2A AMVs ingested in GEOS ADAS per day is about 120,000 and 100,000 , respectively, and 65–75% of these observations are filtered out. For GK2A AMVs, approximately 20–30% of observations are filtered out through QC criteria from seventh to tenth applied to eliminate wind speed bias of GK-2A AMVs among all criteria. Some biases in GK2A still remain in the upper atmospheric layers as shown in Figure 9a. The biases remain as the PD is not large enough to trigger the height assignment error check.

The assimilation of GK2A AMVs has a distinct effect on the mean wind analysis in GK2A's observation area. The analysis difference between the experiment adding GK2A AMVs and the control experiment are the largest over the tropics and southern hemisphere and are up to 2–3 m/s at most levels (not shown). Figure 10 shows differences between GK2AHE and CNTL analysis errors in u-wind at 850 hPa. GEOS analysis are validated against the ERA5 reanalysis of the European Centre for Medium-Range Weather Forecasts (ECMWF) [44] and these errors are absolute differences between GEOS analysis and ERA5 reanalysis. This figure represents closeness of GEOS to ERA5 reanalysis: |GK2AHE–ERA5|–|CNTL–ERA5|. Blue areas indicate that GK2AHE is closer to ERA5; red areas indicate CNTL is closer to ERA5. GK2AHE shows a large reduction in analysis errors in the GK2A's observation area, especially in the tropics. It is due to smaller wind speed biases and RMSVD of the GK2A AMVs compared to the H-8 AMVs in the tropics (see Figure 9).

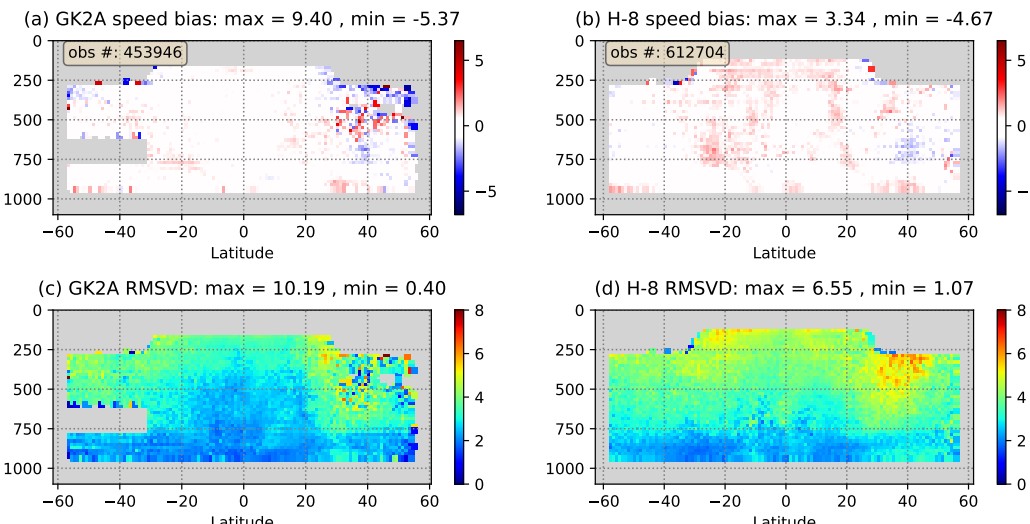

**Figure 9.** Zonal plot of wind speed bias (**a**,**b**) and RMSVD (**c**,**d**) of GK2A IR winds (**a**,**c**) in GK2AHE experiment and H-8 IR winds (**b**,**d**) in CNTL experiment.

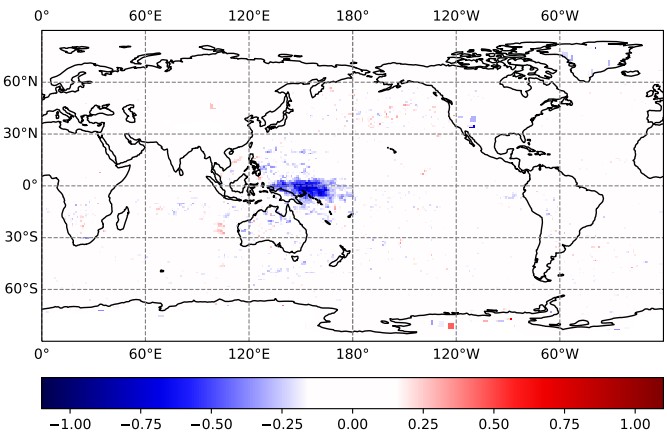

**Figure 10.** Averaged difference between between CNTL and GK2AHE u-wind analysis errors at 850 hPa from 1 December 2020 to 31 January 2021. It represents closeness of GEOS analysis to ERA5 reanalysis: |GK2AHE-ERA5|-|CNTL-ERA5|. The blue color means a reduction in analysis error by adding GK2A AMVs.

Additionally, we compare the fits of radiosonde observations to the model background of the experiments in Table 2 to verify the GK2A AMVs impact in analysis. We take experiment NOH8 in which neither H-8 nor GK2A are assimilated as the initial basis of comparison. Each panel of Figure 11 shows globally–averaged mean differences and ratio of standard deviation for two given experiments. The top row shows results for radiosonde zonal wind and the bottom row for radiosonde temperature; 95% uncertainty is indicated by whiskers for the differences of absolute mean, and by a shaded area for the ratio of standard deviations. Significance for the mean differences relies on z-scores, and significance for the ratio of standard deviations relies on chi-square; both set to 95% confidence [45]. From the perspective of residual statistics, adding extra AMV information from either H-8 or GK2A reduces zonal wind O–B standard deviations, leaving the mean basically unchanged, as shown in Figure 11a,b, respectively. The error reduction from either of these data sources is comparable at low levels, and more noticeable around 250 hPa when AMVs from H-8 are assimilated as opposed to GK2A. When AMVs from both H-8 and GK2A are assimilated (Figure 11c), the mean wind residuals are still unchanged with the standard deviations showing a minor but significant reduction from low levels to about 300 hPa. The effect on zonal wind radiosonde O–B residuals of ultimately adding

GK2A observations to the standard configuration of GEOS ADAS is shown in Figure 11d. Results for the mean are mostly neutral and significantly positive at 200 hPa; the standard deviation ratio shows some statistically significant deterioration of the zonal winds in the layer between 200 and 300 hPa. This is attributed to the large speed biases and RMSVD near the jet stream. The effect on temperature (Figure 11e,f), is essentially neutral in both mean and standard deviation.

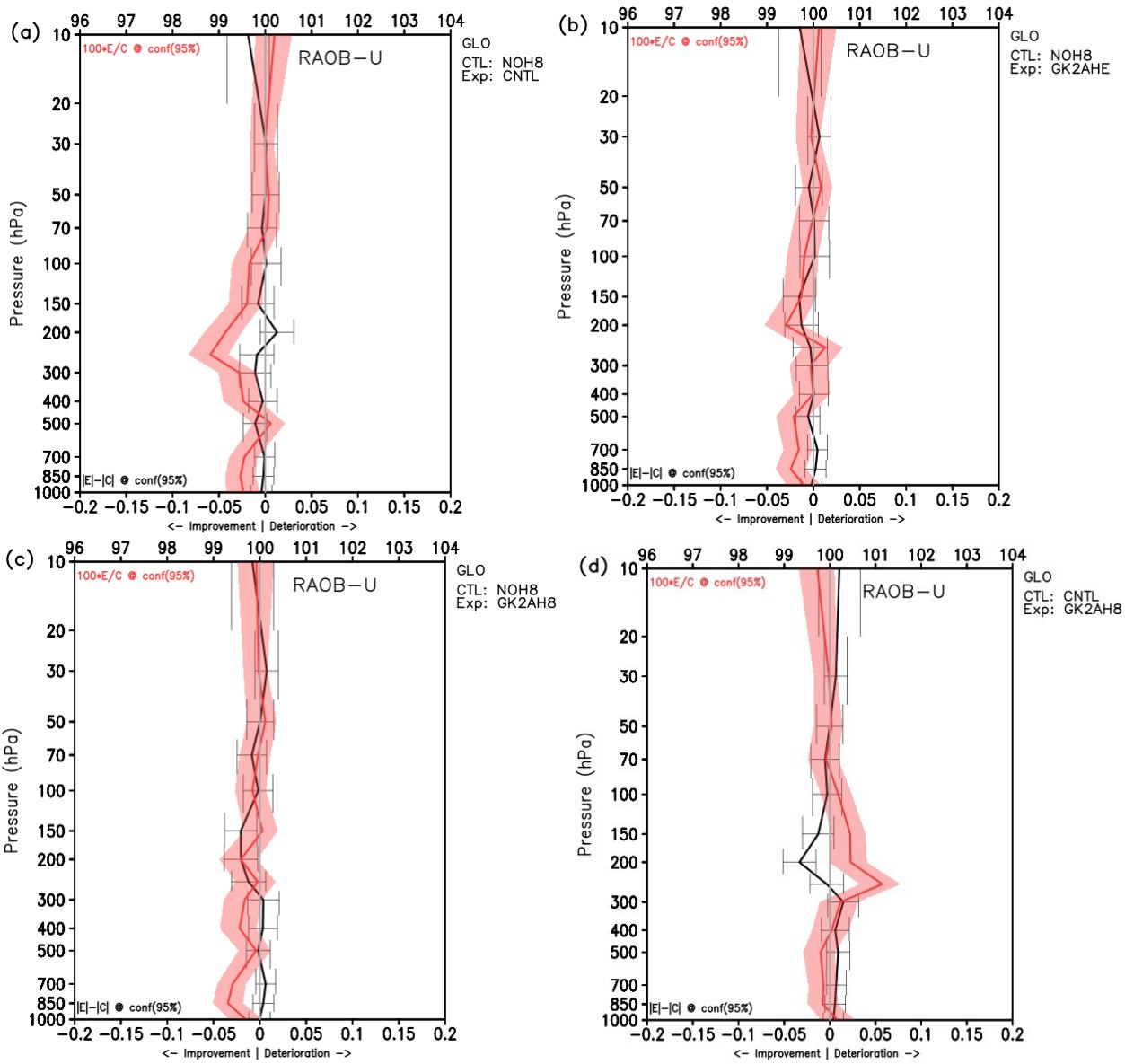

**Figure 11.** *Cont.*

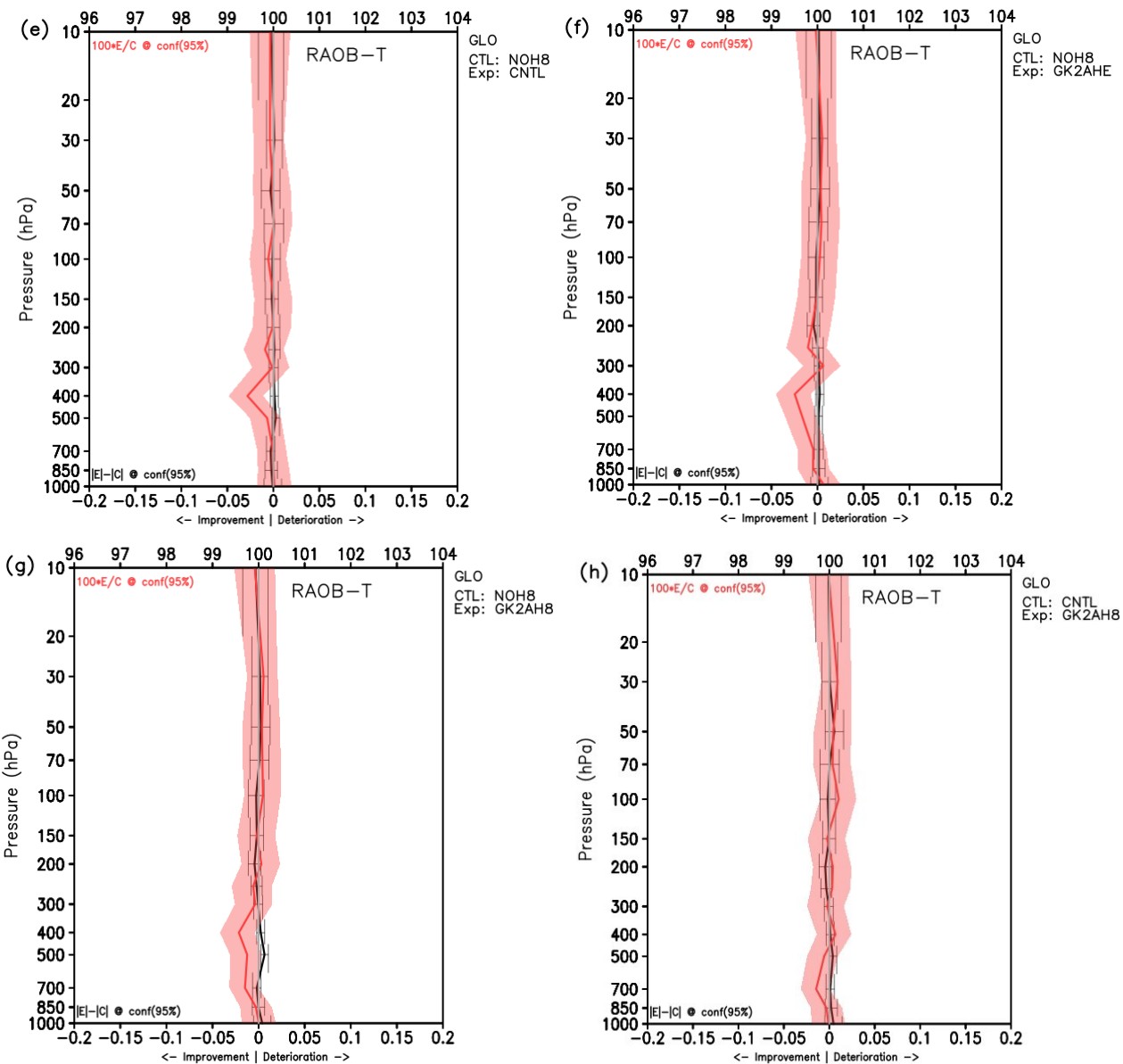

**Figure 11.** Comparison of globally-averaged mean absolute difference (black curve) and standard deviation ratio (red curve) for observation-minus-background fits of radiosonde zonal wind (top row) and temperature (bottom row) between given two experiments: (**a**,**e**) baseline versus addition of H-8; (**b**,**f**) baseline versus addition of GK2A; (**c**,**g**) baseline versus addition of both H-8 and GK2A; and (**d**,**h**) CNTL versus addition of GK2A. Statistical significance at 95% level shown as whiskers for mean difference and red shaded area for standard deviation; range scales for the latter are shown in the top x-axis.

## 3.2. Forecast Impact

### 3.2.1. Verification against Era5 Analysis

The following four figures make use of a scorecard to summarize at a glance the changes in anomaly correlations and root mean square errors from forecasts issued for the experiments in Table 2. The scorecards identify the significance of the changes with marks representing confidence of 99.99, 99 and 95 (%) for various fields, forecast lead-times, and averaging regions. Operational NWP centers, such as the ECMWF, U.K. Met Office, and KMA, use scorecards to quantitatively assess the impact of data assimilation/model changes or new observations in their NWP systems and to determine whether to apply the changes to a new implementation. Improvements in all forecast fields, over all forecast

times and entire regions, are exceptional. Therefore, the changes can be accepted if a certain percentage of improvement is shown in major variables, and the scorecard quantitatively helps with this decision-making. The present study follows a similar procedure to evaluate results from assimilating GK2A AMVs in the GEOS ADAS system. Anomaly correlations and RMS errors can be calculated with respect to an experiment's own analyses or with an independent set of analyses; here, results are calculated using ERA5 reanalysis for verification.

Figure 12 shows a scorecard comparing scores for NOH8 (reference) and CNTL over the period of 1 December 2020 to 31 January 2021. Roughly, green symbols represent improvement (results from experiment being closer to ERA5) while red ones represent degradation (reference being closer to ERA5). CNTL is the current configuration of GEOS, which assimilates H-8 AMVs. The effect of adding H-8 AMVs can be evaluated by comparing the results of CNTL with a baseline experiment NOH8 that uses no H-8 data. Adding H-8 AMVs is seen to bring slightly positive improvements in the northern hemisphere and the tropics and neutral in the southern hemisphere.

**Figure 12.** Scorecard comparing CNTL and NOH8, when both are verified against ERA-5 analyses, and NOH8 is used as reference. This shows the effectiveness of adding H-8 AMVs to a configuration of GEOS that uses no H-8. The three scorecard panels are for Northern Hemisphere (left), Southern Hemisphere (middle) and Tropics (right). Scores are derived for the two month period from 1 December 2020 to 31 January 2021.

Figure 13 shows another scorecard, but now assessing results from adding GK2A AMVs (GK2AHE) to the baseline experiment NOH8. In the absence of H-8 AMVs, assimilation of GK2A AMVs shows considerable improvement, most noticeably in tropical winds. Positive impact is also evident in the mass fields (temperature, pressure, and geopotential height) and the humidity field. Some degradation is seen in the RMS errors of low-level tropical geopotential height and sea level pressure. Unlike in the extra-tropics, the effect of wind changes introduced by assimilating the additional AMVs is only weakly connected with geopotential heights in the tropics. Proper evaluation in the tropics requires the flow to be decomposed into its Rossby and inertia-gravity modes (e.g., [46]); this is left for future work. The sensitivities of low stratospheric humidity to changes in the tropical troposphere are typical in GEOS and are hard to evaluate without a dedicated effort focus on this aspect. Most of the time, these impacts are ignored.

**Figure 13.** Same as in Figure 12, but for comparing the experiment that adds GK2A AMVs (GK2AHE) with the baseline experiment NOH8.

A comparison of Figures 12 and 13 indicates the relative effect of assimilating either H-8 or GK2A AMVs. Assimilation of GK2A AMVs shows larger overall improvement to forecasts in the tropics and southern hemisphere than AMVs from H-8. Furthermore, GK2A AMVs show some improvement after day 2 of forecasts in the northern hemisphere, while H-8 AMVs show some improvement in the early forecast.

A more direct assessment of the impacts of assimilating either H-8 or GK2A AMVs can be seen in Figure 14 in which forecast scores from the present configuration of GEOS, assimilating H-8 AMVs, are compared with results from GK2AHE, the experiment that replaces H-8 with GK2A. Just as concluded from Figures 12 and 13, the two data sets are seen to contribute rather similarly to forecast skills (scorecard is dominated by gray ones), except in the tropics where GK2A is seen to have the upper hand (green symbols indicating slightly larger improvements from assimilating GK2A than H-8). The improvements due to GK2A affect positively tropical temperature regardless of levels; tropical wind and moisture below 500 hPa also seem to benefit slightly more from GK2A; H-8 is seen to contribute slightly more to the early prediction hours of winds at 250 hPa, but the contribution neutralizes after about a day and a half.

Evaluation of how forecast skills are affected by the ultimate configuration that attempts to add GK2A AMVs to the present configuration of GEOS ADAS (i.e., GK2AH8) appears in Figure 15; in this case, the scorecard is built with respect to the CNTL. Although there is a slight degradation in the early part of the forecast in the northern hemisphere, improvements from assimilating GK2A are still manifested in the tropics and even slightly in the southern hemisphere.

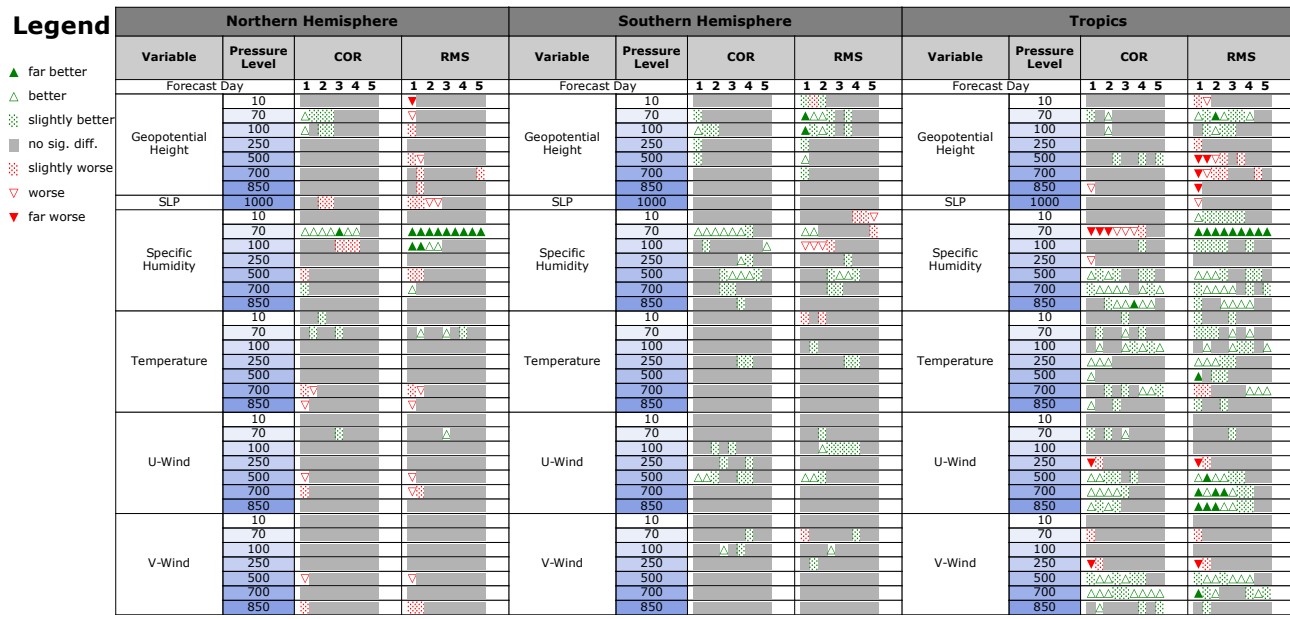

**Figure 14.** Same as in Figure 12, but for GK2AHE (experiment) and CNTL (control) against ERA-5 analysis. This shows the effect of replacing H-8 AMVs with GK2A AMVs in the operational GEOS.

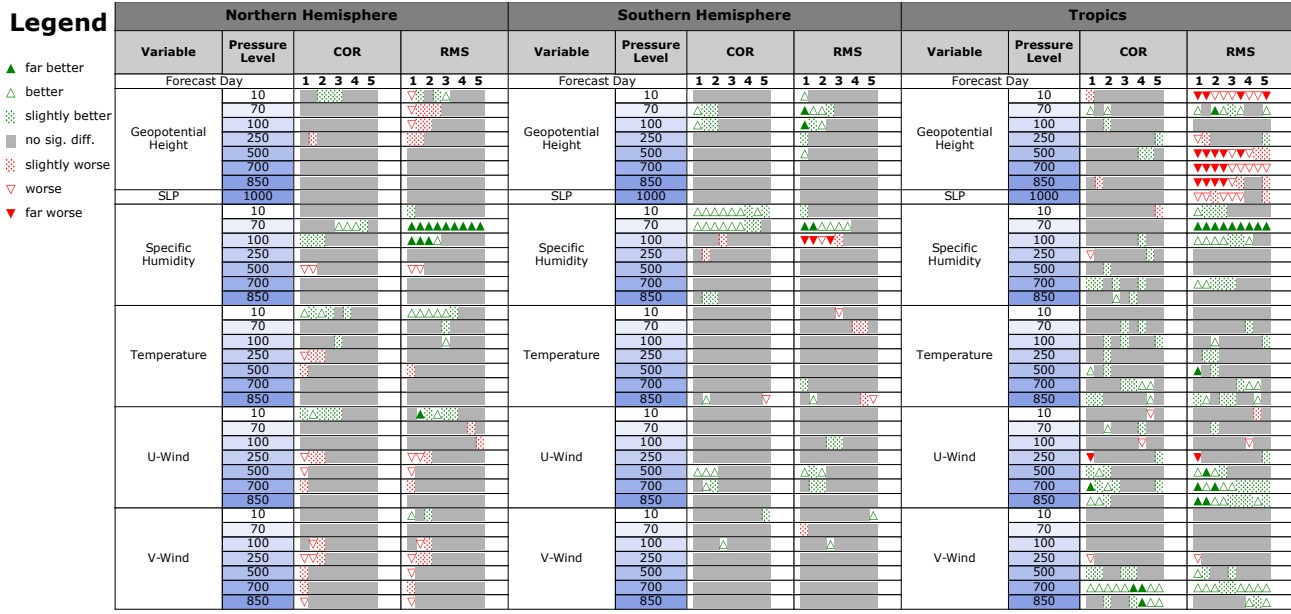

**Figure 15.** Same as in Figure 12, but for comparing the present configuration of GEOS ADAS, which assimilated H-8 AMVs (CNTL), with the experiment GK2AH8 which adds GK2A AMVs to CNTL.

### 3.2.2. Forecast Sensitivity Observation Impact

As a final set of evaluation diagnostics, this section presents results from using the GMAO Forecast Sensitivity Observations Impact (FSOI) tool. The current implementation in GEOS ADAS allows for a *subjective* assessment of how each component of the observing system contributes to the reduction of errors in the 24-h forecasts. This methodology is a combination of the adjoint-based approaches of [47,48]. The assessment is subjective in the sense that forecast error is defined with respect to an experiment's own analysis, and that it is based on a so-called moist-total energy norm (in J kg$^{-1}$) that puts impacts of different observation; the norm gives emphasis to responses in the troposphere. A negative FSOI indicates an observation's contribution to forecast error reduction; a positive value indicates the observation's undesired contribution to forecast error increase. Figure 16 provides a summary of globally averaged AMV FSOI results, calculated from 0000 UTC

forecasts issued in the month of December 2020. The figure compares FSOI results for H-8 and GK2A AMVs assimilated in the ultimate experiment GK2AH8 that adds GK2A to the present configurations of GEOS ADAS. The data count for AMVs from both satellites are comparable, though there are slightly more H-8 observations passing quality control in 300–500 hPa layer, which is attributed to the strict quality control criteria applied to GK2A as described in Section 2.4. Overall, the impact of both datasets on error reduction in the 24-h forecasts are very similar. GK2A shows larger impacts than H-8 in the lower levels whereas H-8 exerts higher influences in the mid tropospheric levels.

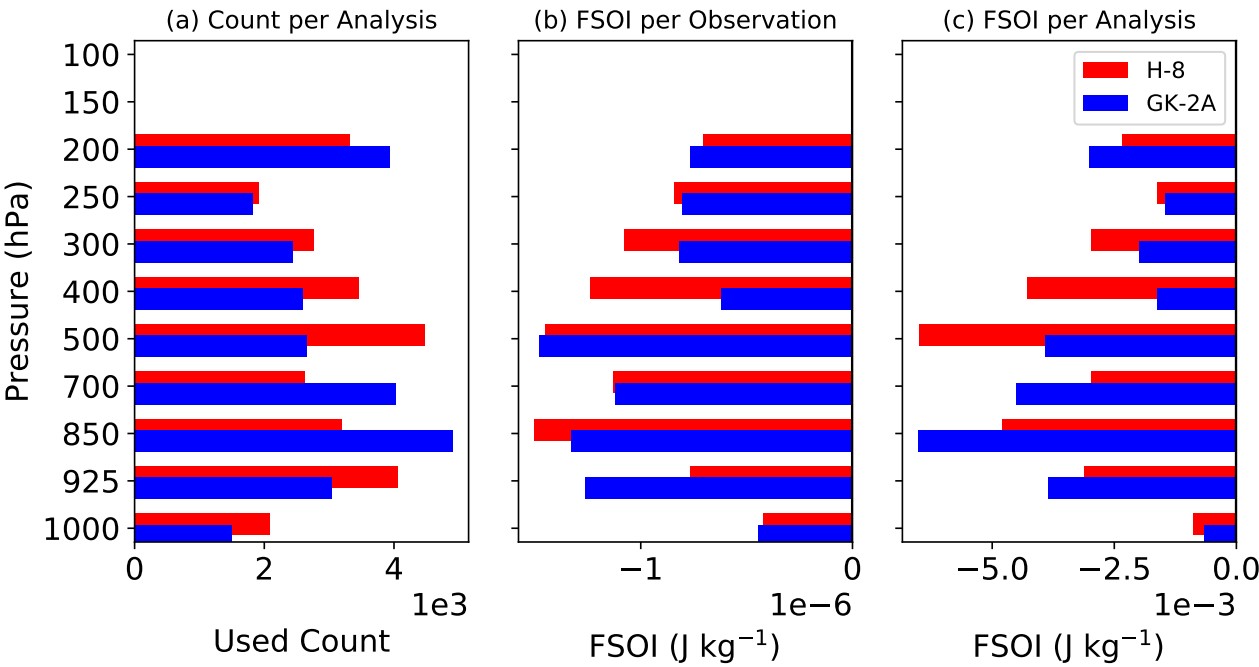

**Figure 16.** Comparison of (**a**) observation count per analysis, (**b**) mean impact per observation and (**c**) mean impact per analysis of GK2A AMVs (blue) and H-8 AMVs (red) in global areas.

A partition of FSOI in tropics, southern hemisphere and northern hemisphere is shown in Figure 17. These largely corroborates results from the scorecard in Figure 15. The impact of GK2A is slightly larger than that of H-8, in the tropics (Figure 17c) and southern hemisphere (Figure 17f) below 500 hPa; above 300 hPa the two contribute similarly; and in the in-between layer, H-8 has larger impact than GK2A. In the northern hemisphere (Figure 17i), H-8 still appears as the one contributing the most to reducing short-term forecast errors as compared to GK2A. The number of assimilated observations (Figure 17a,d,g) from the two satellites is similar except in the northern hemisphere (Figure 17g). This is due to GK2A AMVs, having large speed bias and RMSVD near the northern hemispheric jet stream, and thus being removed along the jet stream. However, when it comes to impact per observations (Figure 17b,e,h) the two data sets show very similar influence in reducing forecast errors, with only very minor exceptions seen.

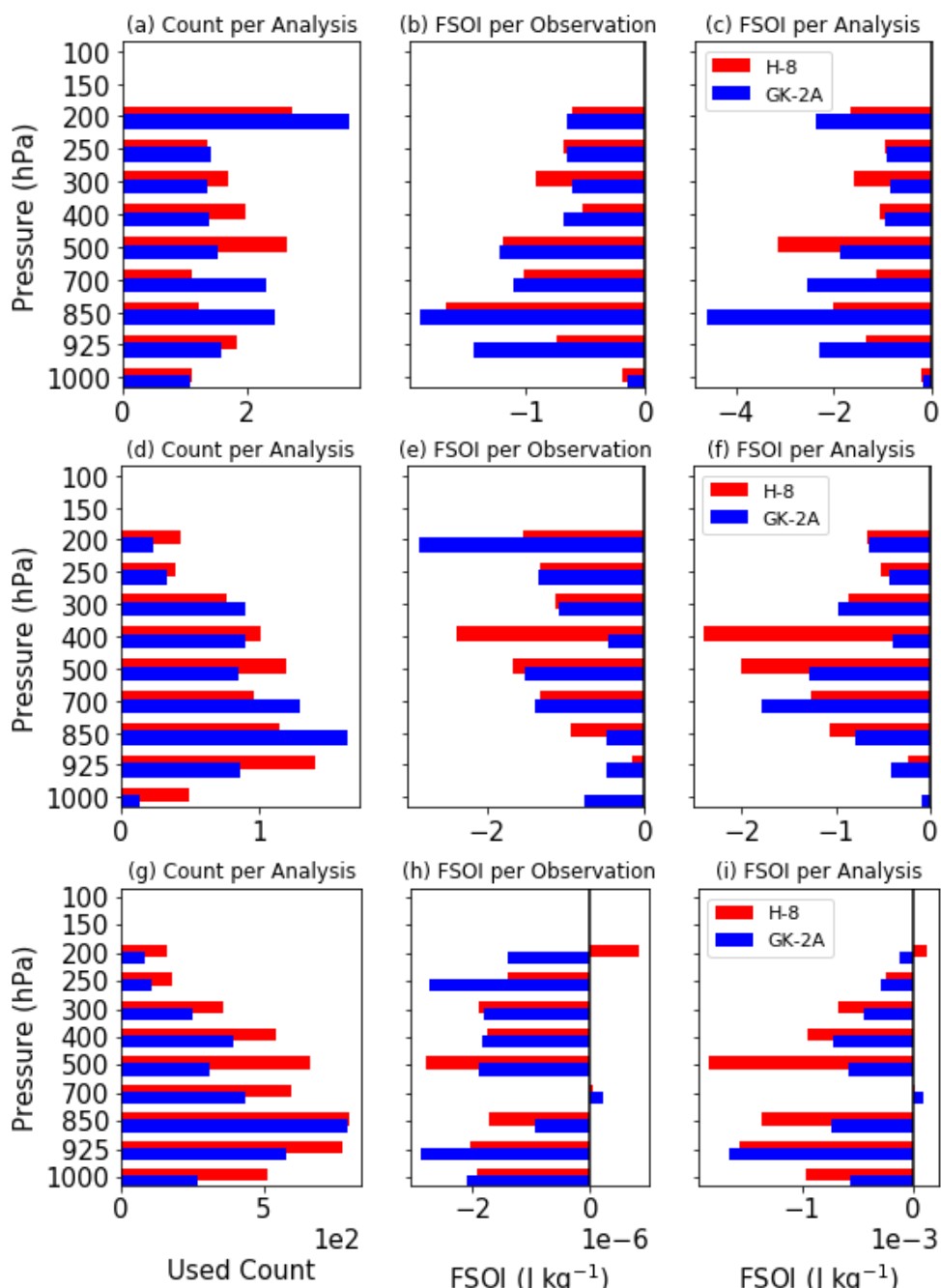

**Figure 17.** Same as Figure 16 except for the Tropics (**a**–**c**), Southern Hemisphere (**d**–**f**) and Northern Hemisphere (**g**–**i**).

## 4. Summary and Discussion

In this study, the quality of GK2A AMVs is evaluated by statistically comparing them with background winds and H-8 AMVs within GEOS ADAS system. GK2A AMVs have similar channel charateristics, resolution and observation area to those of the H-8 AMVs. Since H-8 AMVs are already assimilated in the current version of GEOS ADAS, quality and assimilation impact comparisons between the two satellite data are necessary to assimilate GK2A AMVs in the system. GK2A AMVs have a systematic slow wind speed bias at the lower layer and a relatively large bias and RMSVD at the upper layer during winter in the extratropics. H-8 AMVs show relatively smaller bias and RMSVD in the extratropics, and slight positive bias and larger RMSVD in the tropics. The systematic slow speed bias at low layer is mainly caused by the EBBT height assignment method of GK2A AMVs,

which tends to assign wind vector higher than the model best-fit pressure. The area with large RMSVD and slow speed bias in the upper layers of northern hemisphere is almost identical to the area with large vertical wind shear near the subtropical jet stream in winter. QC criteria are suggested in order to gain maximum benefit from GK2A AMVs based on quality assessment results. A new QC criterion using height error is introduced to remove the large speed bias to avoid negative impacts. After being screened with the QC criteria, the data quality of GK2A AMVs is comparable with the H-8 AMVs. In this study, data with large speed bias are removed for initial impact evaluation, but PD and speed bias were confirmed to have a large correlation, so a further study on the predictors of such bias will be able to calculate the bias correction equation for AMV.

OSEs are conducted to evaluate the effect of the GK2A AMVs on the GEOS ADAS system. GK2A AMVs have an overall positive impact on the GEOS forecasts regardless of the presence or absence of H-8 AMVs, and particularly with a significant improvement of 1.9% in the maximum improvement rate of wind field forecast in the southern hemisphere and the tropics. In addition, FSOI values in the global domain show that the overall contribution to the reduction of forecast errors from GK-2A AMVs is greater than that of H-8 AMVs. Despite the superior quality of the H-8 AMVs, those data have relatively smaller contribution to the reduction of forecast errors in the OSE results and FSOI than GK-2A AMVs, which demonstrates that a careful QC suitable for the characteristics of each satellite data set is essential to obtain the optimal assimilation effect.

The experiment that assimilates GK2A AMVs without H-8 AMVs confirms the benefit of GK2A AMVs even in the northern hemisphere, but the experiment that assimilates two satellite AMVs together shows a slightly negative impact on the short-range forecast time by 2–3 days. Observation redundancy may be the cause since the observation areas of GK2A and H-8 are considerably overlapped and the overall network of observations in the northern hemisphere is relatively dense compared to the southern hemisphere and the tropics. Given observation redundancy may exist between GK2A and H-8, follow-up studies are needed to reduce the correlation between the two satellite data sets while simultaneously taking advantage of the complementary benefits of the two data by assigning optimal observation error and thinning distance, selecting better quality data, and improving the quality of GK2A AMVs. However, the promising results of this study are grounds to encourage the use of GK2A AMVs in NWP systems.

**Author Contributions:** Conceptualization, investigation, methodology, software, validation, and visualization, E.L.; resources, B.M.K.; data curation, A.S.; writing—original draft preparation, E.L.; writing—review and editing, E.L., R.T., J.J., B.M.K. and S.K.P.; supervision, S.K.P. All co-authors discussed with E.L. about concepts, experiments, and results in this study. All authors have read and agreed to the published version of the manuscript.

**Funding:** This study is supported by the National Research Foundation of Korea (NRF) grant funded by the Korea government (MSIT) (NRF-2021R1A2C1095535). It is partly supported under the grant of Basic Science Research Program through the NRF funded by the Ministry of Education (2018R1A6A1A08025520).

**Data Availability Statement:** Data sharing not applicable.

**Acknowledgments:** The first author, Eunhee Lee, was supported by the Korean Government Long-Term Fellowship for Overseas studies. We thank Will McCarty for their insights, support and participation in the early part of the first author's visit to NASA's GMAO. Thanks are also due to the KMA National Meteorological Satellite Center for providing the GK2A AMV observations.

**Conflicts of Interest:** The authors declare no conflict of interest.

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
