# Peer review of "Assessment of Geo-Kompsat-2A Atmospheric Motion Vector Data and Its Assimilation Impact in the GEOS Atmospheric Data Assimilation System"

_remotesensing, doi:10.3390/rs14215287_

Round 1
Reviewer 1 Report
(1) p.3, L.94: Is there a reason why SWIR channel is not used in this study?
(2) p.8, Eq.(1): Have you considered calculating the vertical shear using vector wind difference SQRT[d(u**2)+d(v**2)] instead? This amounts to eliminating the absolute value applied to the wind vector V, i.e. calculating ABS(dV/dz). I believe this approach treats wind as a vector more accurately than Eq.(1). Note hat this suggestion is also in agreement with your Eq.(2), which correctly addresses the wind as a vector.
(3) p.11,L.284-287: Could you elaborate on possibility to introduce AMV bias correction instead of removing some observations through QC?
(4) p.17, L.420-423: Related to previous comment (3), would it be meaningful to combine "good" features of the two AMV methods for improving the bias? They appear to be complementary in some aspects.
Reviewer 2 Report
This paper provides a summary of the quality and impact of atmospheric motion vectors from Geo-Kompsat-2A. It not only characterizes the biases and uncertainty, but also gives explanations and provides a set of rules to form the basis of an appropriate quality control procedure. The observations are shown to add value to the GMAO ADAS system.
While the the broad approach of this study done for similar observations from other satellites, it is essential that this sort of study is published to make the data useful for other centres. I should also note that this has been a very comprehensive study of the data and its impact as well.
I recommend that this be published subject to minor corrections:
1. line 97-98 replace"and the rest channels are 2km" with "and the resolution of the other channels is 2km"
2. line 99: rest --> other
3. line 116: "of forecast consistency" or "for forecast consistency" ?
4. Line 185: "scattering distributions between" --> "scatter plots of"
5. Line 209: "replies" ?
6. Eq (3): which levels is the summation over? All levels or a subset?
7. Line 286. It would be instructive to know what percentage of Himawari-8 winds are routinely filtered as well (also total numbers of winds retained in each case)
8. Figure 9. These plots should be accompanied by number of observations analyzed by the 4DVar. Major differences in observation counts may also account for some of the differences
9. Table 12-15: If Specific Humidity is not to be considered, then it should not be displayed
10. References. Titles of cited articles have a random mixture of capitalizations. Proper nouns such as Meteosat should always be capitalized. Also a few typos (e.g. line533 "frommodel" --> "from model". There maybe others, so suggest thorough and careful check.
Reviewer 3 Report
see report
